# An Investigation of the Behavioral Characteristics of Higher- and Lower-Temperature Group Families in a Condominium Equipped with a HEMS System

**Rajan KC [1],\*, Hom Bahadur Rijal [1], Masanori Shukuya [1] and Kazui Yoshida [2]**

[1] Graduate School of Environmental and Information Studies, Tokyo City University,
Yokohama 224-8551, Japan; rijal@tcu.ac.jp (H.B.R.); shukuya@tcu.ac.jp (M.S.)

[2] Tokyu Real Estate Co.Ltd, Next Generation Engineering Research Center, Tokyo 107-0062, Japan;
kazui_yoshida@tokyu-land.co.jp

\* Correspondence: rkcrajan@gmail.com; Tel.: +81-45-910-2616

**Abstract:** A home energy management system (HEMS) shows the energy used indoors so that the energy waste can be easily identified and reduced. Thermal comfort is related to the trend of energy use in buildings. We conducted a survey in a condominium equipped with a HEMS to determine the indoor thermal environment and various behaviors of the occupants taken for thermal comfort adjustment. The results showed that there is a large variation of indoor air temperatures according to season, floor and flat. We categorized families into two groups, one with higher and the other with lower average indoor temperatures. The indoor air temperature of the higher temperature group in summer was found to be higher than the recommended indoor temperature during the summer season in Japan. The higher temperature group tended to adopt behaviors, such as window opening and using a fan more often, than the lower temperature group. Due to the moderately high insulating levels in the building surveyed, the indoor air temperature of both groups was not low in winter. Heating was used less and irregular. The overall results indicate that the groups of families behaved differently to adjust the indoor thermal environment even though they were equipped with the same HEMS system.

**Keywords:** HEMS; indoor environment; air temperature; relative humidity; occupants' behaviors

## 1. Introduction

Household energy use has been increasing due to the installation of more electric and electronic appliances, such as air conditioning units, computers, electric toilet pans, digital video disc (DVD) players, etc., all of which are basically used to make lifestyles easier and better. Increasing energy use may cause energy scarcity for our future generations. While new types of energy supply, such as fuel cells used for electricity generation, are being developed, they may cause an increase in $CO_2$ emissions. For sustainable energy use, it is important to manage energy use effectively. Seeking an effective way of energy management is important in any building types, but residential buildings in particular, because residential energy use is increasing [1]. The use of smart devices with an effective energy management system is a major concern of developers and buyers in housing corporate sectors, including in Japan.

Home energy management systems (HEMS) are connected through the visualization of electric power use to create a smart home environment and to provide power control of home appliances. Because energy waste can be easily recognized, HEMS should be effective for energy management. The Japanese government aims at establishing HEMS in all new dwellings by 2030 but there is no clear

evidence that HEMS use has reduced energy use in Japan. The number of home electric appliances used determines the electricity use and also influences the thermal environment inside the domestic buildings [2]. The use of HEMS might be useful for reducing energy waste in the number of home appliances being used according to their necessity.

Adaptive behaviors of occupants are not usually taken into consideration in thermal comfort standards, although there might be various adaptive behaviors adopted in dwellings for thermal comfort. Previous studies [3–5] have proved that occupants adopt various adaptive behaviors for thermal comfort adjustments in naturally ventilated buildings. However, occupants living in any type of dwelling have adaptive opportunities. There have been some studies conducted about the adaptive behaviors of occupants in residential buildings but no similar types of studies of smart living with the use of modern electrical devices and energy management systems. Decision makers and politicians have started to look at occupants' behavior in order to reduce energy use. In Germany, the government authorized a bonus for those who could demonstrate less energy use than in the foregoing year [3], demonstrating that occupants' behavior is also more or less concerned with energy saving.

Thermal comfort is "the satisfaction of the mind with the thermal environment and is assessed by subjective evaluation" [6]. Indoor thermal comfort is associated with the trend of energy used in a building. Energy use is associated with the behaviors of the occupants living in those buildings. Studies in the UK and Australia showed that improving people's behavior is considered a useful method to reduce energy use and carbon emissions [4,5].

Mechanical heating and cooling are the major behavioral factors of occupants for regulating indoor thermal comfort, and are the main reasons for indoor energy use. Those living in cold regions may enjoy a sufficiently warm built environment, while, on the other hand, those in temperate regions may not feel warm enough due to a lack of proper heating systems [7]. On the other hand, over-heating or over-cooling may result in an excessive use of energy. There are different guidelines provided for indoor heating and cooling, and for indoor comfort temperature. But existing guidelines in dwellings may be inappropriate, since they do not reflect the effect of occupants' behavior and could thus be more flexible [8]. In the case of free running (FR) mode in both domestic and non-domestic buildings, the range of indoor air temperatures tends to be high because of window opening. On the contrary, it is believed that buildings in which occupants use mechanical heating and cooling to regulate indoor comfort tend to have a narrower temperature range because the indoor environment is well controlled by mechanical means [6]. Nicol [8] showed in his field studies that people flexibly adapted various behaviors to ensure their thermal comfort. In order to achieve thermal comfort by adaptation, humans have developed ways such as opening windows, changing clothes, and having hot or cold drinks. Occupants with opportunities of behavioral adaptation can achieve thermal comfort at relatively higher and lower indoor air temperatures in summer and winter than the temperatures by the predicted mean vote (PMV) model [9]. Brager and de Dear [10] showed in their results that the predicted thermal sensation by PMV is warmer than that actually felt by occupants in naturally ventilated buildings.

It is necessary to address the adaptive opportunities of occupants while proposing any guidelines and standards. According to previous studies, if the occupants have more behavioral choices, then the range of temperatures may be made wider than in conventional guidelines. Even in mechanically heated and cooled buildings, there may be a wide range of indoor conditions due to a variety of adaptive behaviors, and in similar types of buildings, different ways of living and adaptive behaviors of families may affect the indoor thermal environment.

In this study, we examined the indoor thermal environment of families living in a single condominium equipped with a HEMS. We studied the indoor air temperature variation of all the families throughout one year. We found a large variation of indoor air temperatures, not only according to seasons, but also according to the families, and that occupants do not necessarily adopt similar behaviors according to season, floor or flat. Hence, we categorized families into two groups, one with higher and the other with lower average indoor air temperature, and studied the behavior of the

two groups in summer and winter, and how they adjusted thermal comfort despite the temperature differences. The findings should be useful for the improvement of HEMS or the new development of HEMS in the future.

## 2. Methodology

### 2.1. Studied Area and Building

Our study site was Shinagawa, which is located in the southern part of the Tokyo metropolitan area. This area has a warm and temperate climate. There is significant rainfall throughout the year [11]. Annual average precipitation is 1469 mm. The warmest month of the year is August with a 31.6 °C mean maximum temperature; January has a 1.8 °C mean minimum temperature. The mean maximum relative humidity is 78% in August and 55% in January. The mean maximum, average and minimum outdoor and indoor air temperature variations along with the relative humidity in different months are explained in Section 3.

An 18-storey condominium [12] housing 356 families (Figure 1a) was selected for this study. The studied flats in the condominium are mostly 3 bedrooms, a living room and a dining room with kitchen and few are 4 bed room flats. The area of the flats varies from 71 to 90 m$^2$. Figure 1b shows the floor plan of a representative flat.

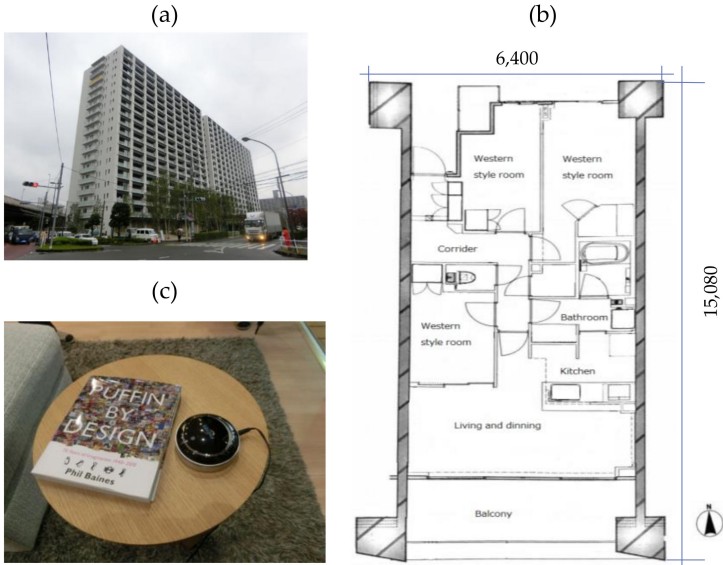

**Figure 1.** Studied condominium with the measuring device: (**a**) Studied building, (**b**) Floor plan of one of the selected flats, and (**c**) Measurement device.

This condominium is made of a reinforced concrete (RC) structure and was completed in 2015. Sprayed insulation of 40-mm urethane on 180-mm-thick RC walls is used with double-glazed windows. KC et al. [12] describe the detailed characteristics of the studied condominium. This condominium is certified as "less carbon architecture" (little CO$_2$ emission and energy saving) according to Japanese building supplementary codes. The studied building is equipped with a HEMS and a compact co-generation system called ENE-FARM. This is a fuel-cell co-generation system that produces electricity through a chemical reaction between oxygen within the surrounding air and hydrogen extracted from natural gas [13], and the heat produced during this process is used for heating water to be used in kitchens and bathrooms. The specific co-generation system developed for this condominium was installed in each dwelling unit of the studied building. The installation in this condominium has been claimed as being used for the first time in Japan. There are no high-rise buildings surrounding this condominium so that

natural ventilation is able to perform well. There is a green environment around the building and a river flows to the east. The surrounding environment in general is very good.

### 2.2. Thermal Environment Measurement

The device shown in Figure 1c was used for measurement and recording of indoor air temperature and relative humidity. The data was recorded in the interval of 2–10 min. The device has a weight of 240 g. The accuracy of sensors for measurement of air temperature was $\pm$ 2 °C in the range of 0–40 °C and $\pm$ 5% for relative humidity in the range of 0–100%. The accuracy of the measurement sensors was slightly low, but allowed for inexpensive installation used at large scales. The devices were provided to each flat in the condominium, so it was necessary to consider the cost of the devices. We calibrated the sensors used in the condominium with the high accuracy sensors (thermistor with accuracy of $\pm$ 0.5 °C and $\pm$ 0.3 °C for air temperature and polymer-membrane with the accuracy of $\pm$ 5% for relative humidity) [12]. All the data were arranged using the regression equation derived from the relationship of the data obtained from calibrated devices and the actual used devices. The measured data was matched with the voting times of the occupants especially for analyzing their behavior. The outdoor air temperature was taken from the Tokyo Meteorological Station located in Chiyoda district, which is the nearest meteorological station at almost 13 km away from the studied building. The outdoor air temperature was provided at an interval of 10 min. For uniformity, the indoor measured data was converted to a 10-min interval. Although the device was installed in each flat, only 41 flats had continuous measurement for the whole year, so the data of these 41 flats was analyzed to understand the indoor thermal environment.

### 2.3. Occupant Behavior Survey

An online survey was conducted to observe the level of thermal comfort and the occupants' adaptive behavior. The occupants were asked a series of questions on various behaviors taken to adjust their thermal environment [12]. To analyze adapted behaviors like window opening, fan use and the use of heating or cooling, binary data (0 = Off, 1 = On) were collected. The clothing insulation value was listed as shown in Figure 2. Clothing insulation values have been estimated based on OM Solar Japan, similar to a previous study [14], and occupants were requested to choose the clothing insulation value that fitted what they wore at the time of voting. The purpose of the survey was explained to occupants in advance. It was explained that the purpose of the data will be used for statistical analysis only. Altogether, 33 males and 37 females of 47 families voted 17,026 times during the survey period from November 2015 to October 2016. The ages ranged from 15 to 75 years. The age groups were categorized with an interval of 5 years; the largest number of votes was received from the groups of 35–39 years, 50–54 years and 70–74 years. The data was categorized into free running (FR), which meant occupants had not used heating and cooling at the time of voting, cooling (CL), which meant occupants were using cooling devices, and heating (HT), which meant occupants were using heating devices. MM mode stands for the condition when there is no separation of FR, CL or HT modes.

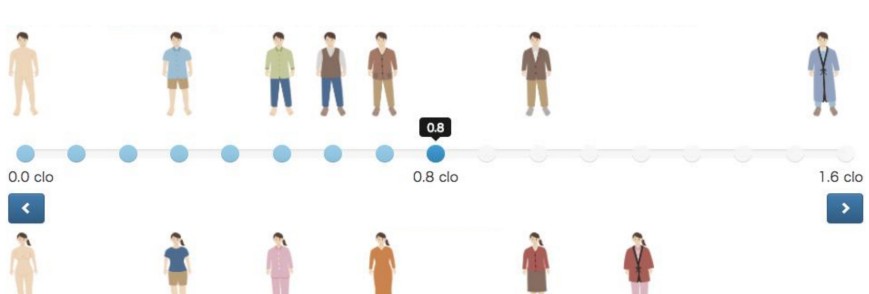

**Figure 2.** Questionnaire plate for clothing insulation value (actual questionnaire was provided in Japanese) [12].

## 3. Results

### 3.1. Thermal Environment

#### 3.1.1. Monthly Indoor and Outdoor Air Temperature

The relationship between monthly mean indoor air temperature of continuous measured data and monthly mean outdoor air temperatures was investigated to understand how the indoor air temperature fluctuated corresponding to outdoor air temperature. As shown in Figure 3, indoor air temperature fluctuated in accordance with outdoor air temperature variation but the amplitude was smaller. Realized indoor air temperatures were different from one month to another. The relative differences between indoor and outdoor air temperatures were greater in January and smaller in August. The indoor air temperature was almost 20 °C for the months with monthly outdoor air temperature below 10 °C; that is, in January, February and March. The highly insulating materials used in the building were one of the reasons for this temperature difference other than heating use, because similar trends were observed with flats using no heating use, i.e., in FR mode. The result showed that the indoor temperature changed with the months. Similar results have been obtained from studies of residential buildings in China and the UK [7,15].

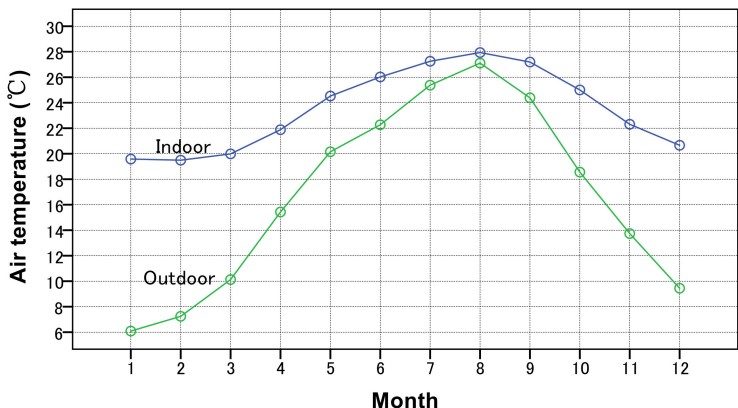

**Figure 3.** Monthly mean indoor and outdoor air temperatures.

Figure 4 shows the relationship of indoor and outdoor air temperatures in four respective seasons. December to February is winter, March to May is spring, June to August is summer and September to November is autumn. The indoor air temperature in spring and autumn tends to be highly correlated with the outdoor air temperature. In winter, the indoor and outdoor air temperature difference is quite large; the indoor air temperature is much higher than the outdoor air temperature. This is due to the highly insulating materials used in the building along with the use of heating as mentioned above. In summer, the difference between the indoor and outdoor air temperature is quite small; this indicates that cooling use is not high or occupants might have used a high temperature setting. The linear regression equations for respective seasons are as follows:

$$\text{All } T_i = 0.377T_o + 17.2 \left( n = 1,047,900, \text{ R}^2 = 0.72, \text{ S.E.} = 0.008, \text{ p} < 0.001 \right) \tag{1}$$

$$\text{Winter } T_i = 0.196T_o + 20.2 \left( n = 78,775, \text{ R}^2 = 0.09, \text{ S.E.} = 0.002, \text{ p} < 0.001 \right) \tag{2}$$

$$\text{Spring } T_i = 0.334T_o + 19 \left( n = 82,112, \text{ R}^2 = 0.48, \text{ S.E.} = 0.001, \text{ p} < 0.001 \right) \tag{3}$$

$$\text{Summer } T_i = 0.236T_o + 23.6 \left( n = 804,062, \text{ R}^2 = 0.28, \text{ S.E.} = 0.001, \text{ p} < 0.001 \right) \tag{4}$$

$$\text{Autumn } T_i = 0.391T_o + 19.74 \left( n = 82,951, \text{ R}^2 = 0.66, \text{ S.E.} = 0.001, \text{ p} < 0.001 \right) \tag{5}$$

where, $T_i$ is indoor air temperature (°C); $T_o$ is outdoor air temperature (°C); n is number of data points; $R^2$ is the coefficient of determination; S.E. is standard error of the regression coefficient; and p is the level of significance for the regression coefficient. Table 1 shows the comparison of the regression equations of this study with other studies in the Tokyo and Yokohama areas in Japan. The regression coefficient of this study for winter is lower than others in heating, ventilation and air conditioning (HVAC) and HT modes. Similarly, the regression coefficient of this study for summer is also lower than other similar types of studies conducted in ordinary buildings. The reason might be the effect of heat capacity enhanced by the use of the highly insulating materials in this building compared to other buildings, which were mainly detached houses.

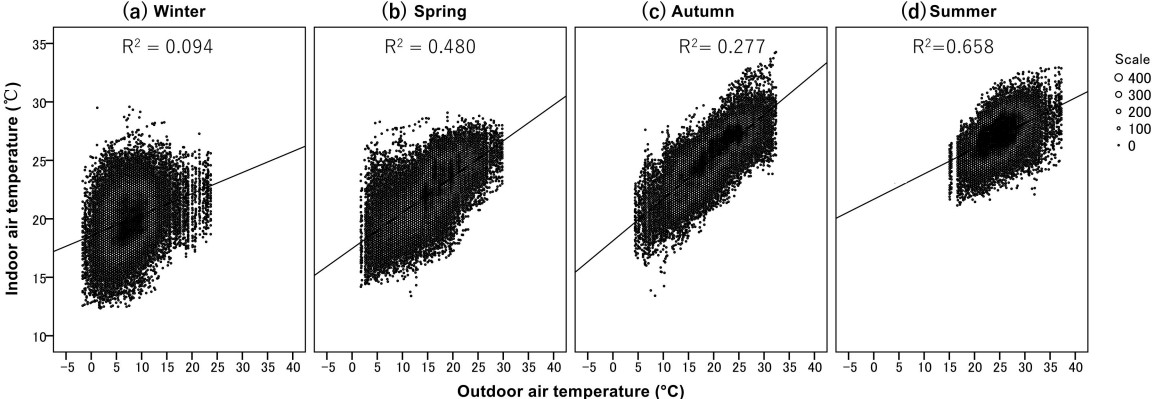

**Figure 4.** The relationship of indoor and outdoor air temperature in each season: (**a**) Winter, (**b**) Spring, (**c**) Autumn and (**d**) Summer.

**Table 1.** Comparison with other studies.

| References | Types of House | Mode | Season | Areas | Equations |
|---|---|---|---|---|---|
| This study | HEMS condominium | MM | Winter | Tokyo | $T_i = 0.196T_o + 20.2$ |
| This study | HEMS condominium | MM | Spring | Tokyo | $T_i = 0.334T_o + 19$ |
| This study | HEMS condominium | MM | Autumn | Tokyo | $T_i = 0.391T_o + 19.74$ |
| This study | HEMS condominium | MM | Summer | Tokyo | $T_i = 0.236T_o + 23.6$ |
| Imagawa et al. (2015) [14] | Ordinary house | FR | All | Tokyo, Kanagawa, Chiba | $T_i = 0.727T_o + 9.4$ |
| Imagawa et al. (2015) [14] | Ordinary house | CL | All | Tokyo, Kanagawa, Chiba | $T_i = 0.257T_o + 20.3$ |
| Imagawa et al. (2015) [14] | Ordinary house | HT | All | Tokyo, Kanagawa, Chiba | $T_i = 0.327T_o + 14.0$ |
| Katsuno et al. (2012) [16] | Ordinary house | HVAC | All | Tokyo/Yokohama | $T_i = 0.230T_o + 21.6$ |
| Katsuno et al. (2012) [16] | Ordinary house | FR | All | Tokyo/Yokohama | $T_i = 0.572T_o + 13.7$ |
| Rijal H.B. (2014) [17] | Ordinary house | FR | All | Kanto region | $T_i = 0.587T_o + 12.6$ |
| Rijal H.B. (2014) [17] | Ordinary house | CL | All | Kanto region | $T_i = 0.183T_o + 22.3$ |
| Rijal H.B. (2014) [17] | Ordinary house | HT | All | Kanto region | $T_i = 0.220T_o + 17.4$ |

### 3.1.2. Indoor Air Temperature by Flats

The indoor air temperature variation of respective flats was analyzed from the measured data to know whether the indoor environment was similar or different depending on families. As 356 families were living in the same building equipped with a HEMS, it can be assumed that a similar indoor thermal environment was realized in all flats in general. Figure 5 shows a large range of variation in indoor air temperature. In winter, the highest difference in indoor air temperature among the flats was 12 °C. This range was smaller in summer, being 6 °C. It is considered that the individual behavior of each family caused these differences in indoor air temperature. The mean indoor air temperature was 19.9 °C in winter, which is quite similar to the 19.5 °C found in a study of English homes [15]. The mean indoor air temperature in summer was 27.1 °C, which is quite similar to the recommended temperature for summer in Japan. The indoor air temperature in summer was observed to be high

for most of flats, which proved that the use of mechanical cooling was not regular. We discuss the proportion of mechanical heating and cooling use of some higher and lower temperature groups in Section 3.2.

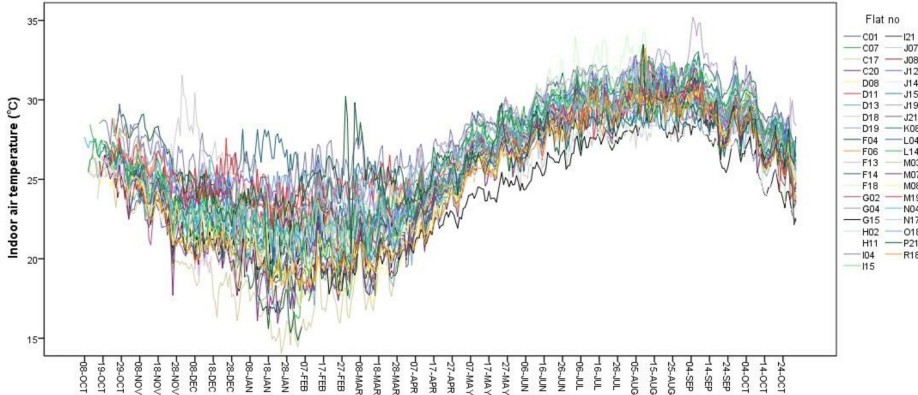

**Figure 5.** Indoor air temperature according to the flats.

For the purpose of understanding the indoor air temperature variation of different families in more detail, we observed one-day indoor air temperature variation of different flats in summer and winter. As shown in Figure 6a, the largest difference between the flats maintaining the highest and the lowest temperature was 10 °C. In all flats, indoor temperature decreased early in the morning and then gradually increased from 07:00 until 14:00–15:00. This was due to solar heat gain and the use of heating. The indoor temperature gradually decreased in the evening period. At around 20:00 onward, the indoor air temperature increased again in some of the flats definitely due to heating use. The heating pattern of some of the flats is quite similar to the study of English homes [15]. As shown in Figure 6b, cooling use did not appear much except in some flats because the fluctuation of the temperature did not sharply decrease. The largest temperature difference between the flats maintaining the highest and the lowest temperature was 6 °C. At 20:00 onward, the indoor air temperature seemed to decrease slightly, mostly due to the use of cooling.

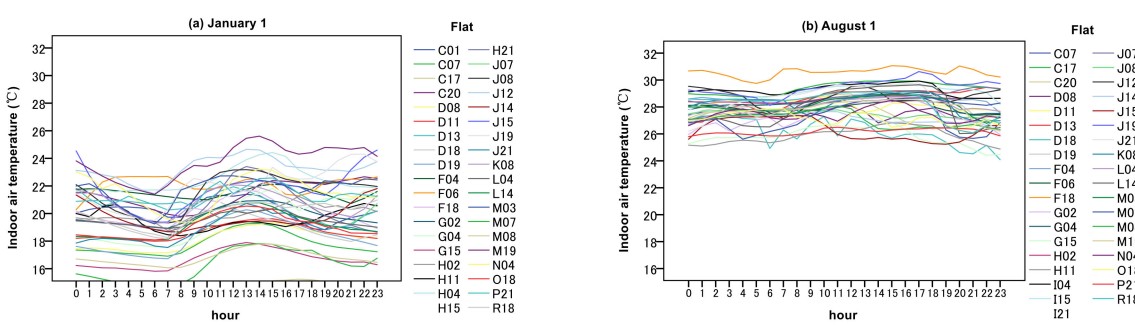

**Figure 6.** Indoor air temperatures: (**a**) January 1 and (**b**) August 1.

### 3.1.3. Seasonal Differences in Indoor Air Temperature by Floor

We observed the mean indoor air temperature of four peak months in respective seasons of every floor in order to understand the effect of floor height from the ground. We had expected that the mean air temperature of the floor must change as the height of the floor changed, but it was not so. As shown in Figure 7, the mean indoor air temperature ranged from 19 to 30 °C in different seasons in different floors. The indoor and outdoor air temperature difference of all floors was large in January, April and October. This is probably due to the mainly effect of the insulating materials used in the buildings. The indoor and outdoor air temperature difference of all floors was quite small in August, probably due to less use of cooling. The seasonal changes of indoor air temperature do not necessarily

look consistent with each other. For example, the indoor air temperature of floor 15 had the highest temperature among all floors in January, but floor 9 had the highest temperature among all floors in August. This suggests that the indoor environment was influenced by occupants' behavior rather than the floor level in the studied condominium since the window orientations of flats were identical.

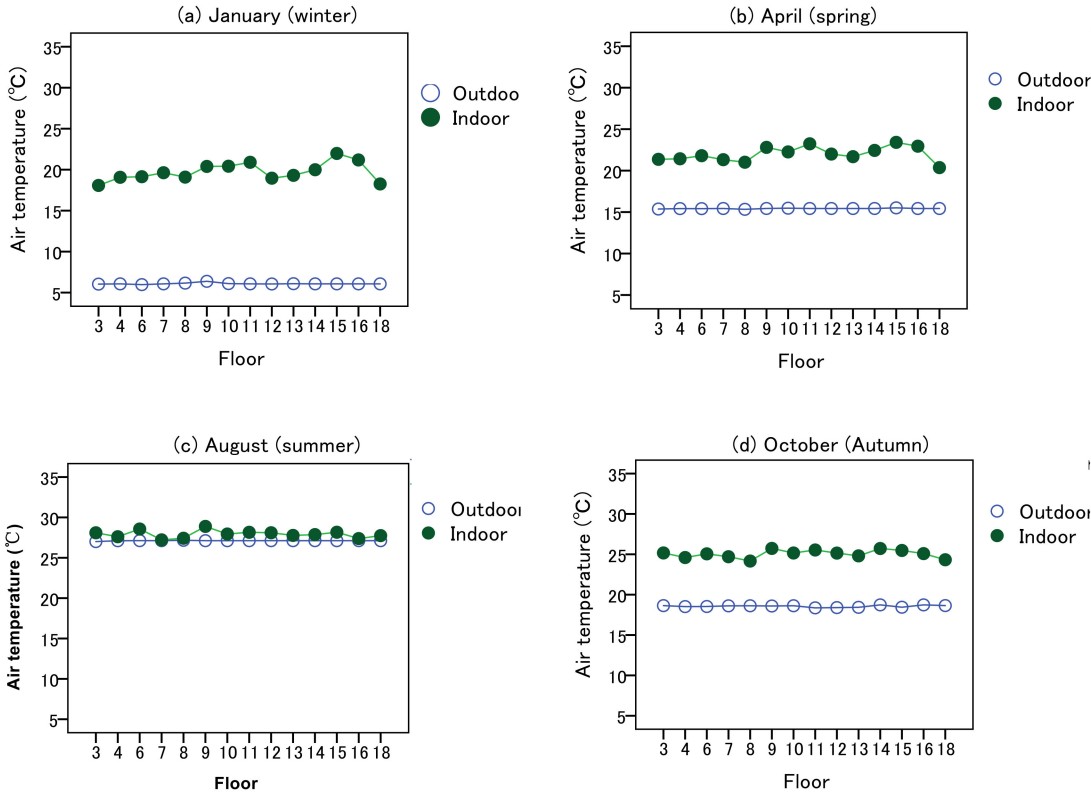

**Figure 7.** Indoor air temperature representative months of all seasons: (**a**) Winter, (**b**) Spring, (**c**) Summer, (**d**) Autumn.

### 3.1.4. Indoor Air Temperature Differences of Two Groups of Families

In Sections 3–3, we noted that the indoor air temperatures were different according to the season, flat and floor. We wondered how people adapted to these temperature differences and what behaviors were adopted for restoring thermal comfort. We listed only those flats with both measurement data and questionnaire data and divided them into four groups, SH and SL (high and low temperature groups in summer), and WH and WL (high and low temperature groups in winter), on the basis of mean indoor air temperature. Table 2 shows mean indoor air temperatures of these groups in summer and winter, with the standard deviation, in two modes: mix mode (MM) and free running (FR). The difference between SH and SL is 1.8 °C in MM mode and 1.7 °C in FR mode. This is due to different individual cooling behaviors in MM mode. The temperature difference in FR mode is due to the difference in behaviors like window or door opening. Similarly, there is 1 °C difference in MM mode and 2.6 °C difference in FR mode between WH and WL in winter. These temperature differences suggest that individual occupants' behavior plays a key role in determining the indoor thermal environment.

We also observed the cumulative percentage of the four groups for higher and lower indoor air temperatures in summer and winter in MM mode. Figure 8 shows the cumulative percentage of indoor air temperature in both higher- and lower- indoor air temperature groups in summer and winter. In the SL group, the amount of time that the indoor air temperature was higher than 28 °C was very small, while, on the other hand, in the SH group it was almost 30%. Similarly, in the WL group, the amount of time the indoor air temperature was higher than 28 °C was small, but in WH group it was more

than 30%. We found 2 °C difference in indoor air temperature between SH and SL. The indoor air temperature difference between WH and WL was almost 4 °C.

**Table 2.** Temperature differences of the groups in different modes.

| Season | Group | Mode | N | Mean $T_i$ (°C) | S.D. (°C) |
|---|---|---|---|---|---|
| Summer | SH | MM | 1276 | 27.9 | 1.2 |
|  | SH | FR | 900 | 27.7 | 1.2 |
|  | SL | MM | 840 | 26.1 | 1.2 |
|  | SL | FR | 720 | 26.0 | 1.2 |
| Winter | WH | MM | 1140 | 22.4 | 1.8 |
|  | WH | FR | 211 | 22.1 | 2.0 |
|  | WL | MM | 1140 | 21.4 | 1.8 |
|  | WL | FR | 660 | 19.5 | 1.5 |

$T_i$: Mean indoor air temperature, S.D.: Standard Deviation.

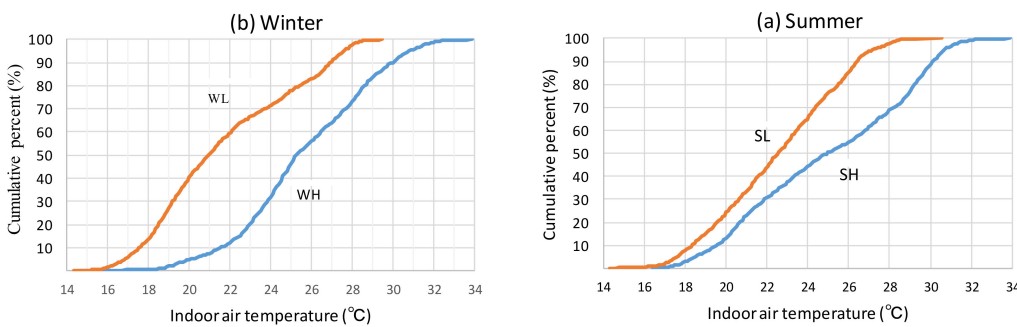

**Figure 8.** Indoor air temperature of higher and lower groups: (**a**) Summer and (**b**) Winter.

### 3.2. Behavioural Differences of the Groups for Thermal Comfort Adjustments

Previous adaptive studies showed that indoor comfort temperature is correlated with outdoor air temperature [10,18–20]. The cause of such dependence of comfort temperature on outdoor air temperature might be either physiological, psychological, behavioral adaptation or some combination of the three. Thermal adaptive behaviors are generally considered related to naturally ventilated or free-running mode, but even in mechanical heating and cooling modes, there may be some adaptive activities [18,21]. In this study, we investigated how behavioral adaptation takes place in smart living and how occupants live comfortably with high indoor air temperatures. Behavioral adaptation includes both passive and active adjustments. Passive adjustments include the use of passive means, such as a change of clothing insulation, window opening, and so on, whereas active adjustments include the use of mechanical heating and cooling devices. Here, we focus on the behaviors of (1) clothing adjustment, (2) door and window opening, (3) fan use, and (4) the use of air-conditioning units. These behavioral characteristics of the occupants are analyzed using logistic analysis method used by Nicol and Humphreys [18] based on the information provided by the occupants during the time of voting. The logistic equations are given with the behavioral characteristics of both higher and lower temperature groups for summer and winter.

#### 3.2.1. Clothing Adjustments

In any dwelling there might be clothing insulation differences due to individual perception of the need for thermal comfort adjustments. Here, we tried to evaluate the clothing insulation differences of the groups maintaining high and low temperatures in summer and winter. Figure 9a shows the clothing insulation differences of SH and SL in summer in FR mode. The occupants of both groups were observed adjusting thermal comfort with low clothing insulation in summer. Comparatively, SH has slightly lower clothing insulation than SL. Figure 9b shows the clothing insulation differences

of WH and WL in winter in FR mode. Interestingly, the members of WH had slightly higher clothing insulation than WL. The WH group might prefer higher temperatures and need more heating if they used low clothing insulation, but the difference in clothing insulation between the high and low temperature groups is not so large.

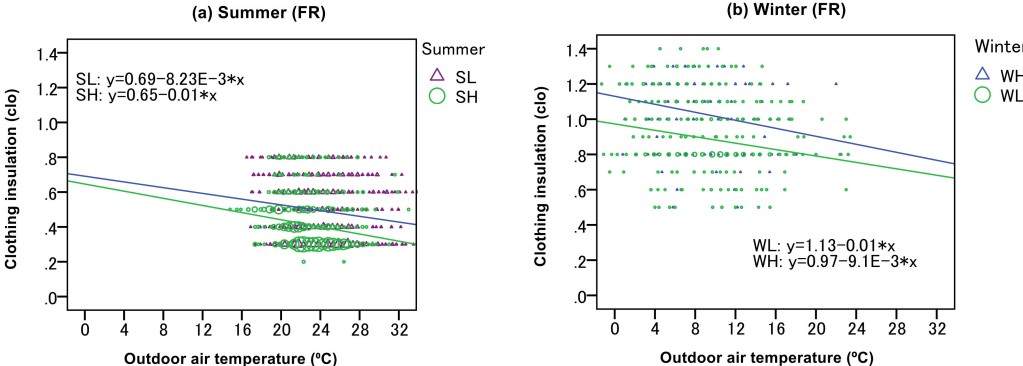

**Figure 9.** Clothing insulation and outdoor air temperature for higher and lower group: (**a**) Summer and (**b**) Winter.

### 3.2.2. Internal Door and Window Opening

Indoor thermal comfort and indoor air quality is mainly affected by the ventilation rate. When windows and internal doors are opened, ventilation takes place and helps to improve the indoor thermal environment. A study in Hong Kong [22] showed that most of people felt stuffy because of poor indoor air quality. Although we did not measure the air velocity during the survey, we took the respondents' votes for window and internal door openings. The measurement of the indoor thermal environment was conducted in the living room, suggesting that the opening and closing of an internal door may play an important role in determining the indoor air quality and indoor air temperature. Figure 10a shows that the proportion of internal door openings for SL is higher than SH in summer. Possibly, it indicates that for the SL group, ventilation was good, resulting in lower indoor air temperatures in FR mode. The window opening behavior gradually increases as the outdoor air temperature increases but the proportion of window openings for SH is lower than for SL. At 25 °C, the proportion of window openings was 0.70 for SL but just 0.30 for SH. Natural ventilation from window opening was good, helping occupants remain comfortable with rather high indoor air temperatures together with less clothing insulation. Humphreys and Nicol [23,24] suggested that the comfort temperature can be increased by 3 °C or 4 °C per 1 m/s increase of air velocity. Thus, it is probable that occupants living in this condominium are comfortable even with high temperatures in FR mode due to a well-ventilated system.

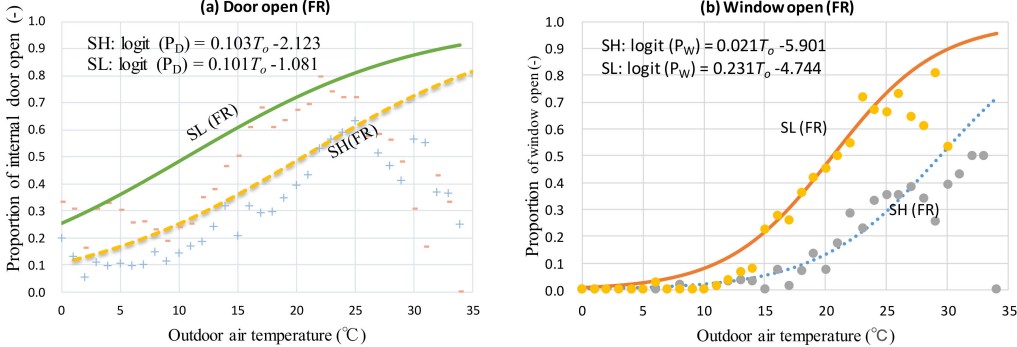

**Figure 10.** Relationship between door or window opening and outdoor air temperature of higher and lower groups: (**a**) Door open and (**b**) Window open.

### 3.2.3. Fan and Cooling Use

Figure 11 shows that the fan use increased along with outdoor air temperature for both SH and SL groups. The occupants used fans rather than air conditioning units for adjusting the thermal environment when the outdoor air temperature was sufficiently low. The proportion of fan use was similar to each other between the SH and SL groups. When the outdoor air temperature was above 29 °C, the proportion of fan use sharply increased for both SH and SL. The use of cooling increased with high outdoor air temperatures, which shows that occupants used cooling only with high outdoor air temperatures. The occupants used air conditioning units for cooling when the outdoor air temperature rose above 25 °C but the rate of increase was not large up to 26 °C for both SL and SH. At high temperatures, the cooling use for SH was less than SL. The proportion of cooling use for SL reached 0.30 at an outdoor air temperature of 27 °C. This is about 0.20 lower than the result of a previous study done in the Gifu area of Japan, which was 0.50 [25]. This might be due to the mean radiant temperature in this condominium being lower than that of the detached houses with rather low insulation levels in the Gifu area, as suggested by human body exergy research [26,27].

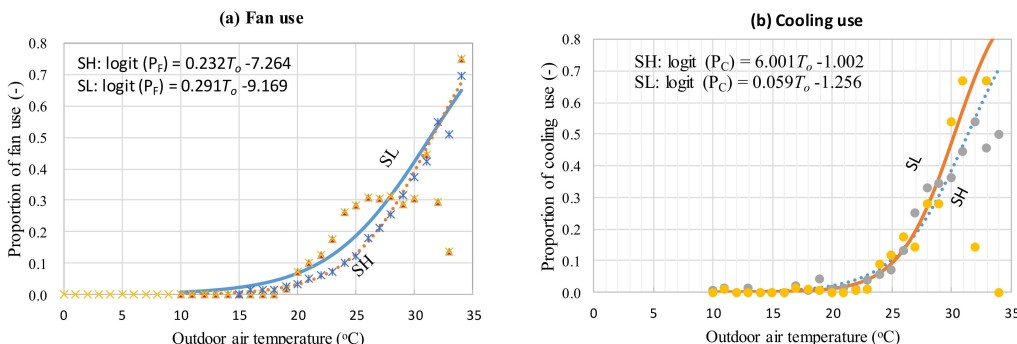

**Figure 11.** Relationship between fan use or cooling use and outdoor air temperature for higher and lower groups in summer: (**a**) Fan use and (**b**) Cooling use.

### 3.2.4. Heating Use of High and Low Temperature Groups in Winter

Figure 12a shows the proportion of heating use for WH and WL groups. The proportion of heating use for both WH and WL gradually increased as the outdoor air temperature fell below 20 °C. The proportion of heating use for WH was higher than WL. The proportion of heating use for WH was 0.85 when the outdoor air temperature was 0 °C, but only 0.29 for WL. The trend of heating use for WH was similar to the result of a study of detached houses of the Tokyo, Yokohama and Chiba areas of Japan, which was 0.90 [14], but the proportion of heating use for WL was much lower. This is probably due to a better level of thermal insulation in this condominium compared to that of other Japanese condominiums and detached houses.

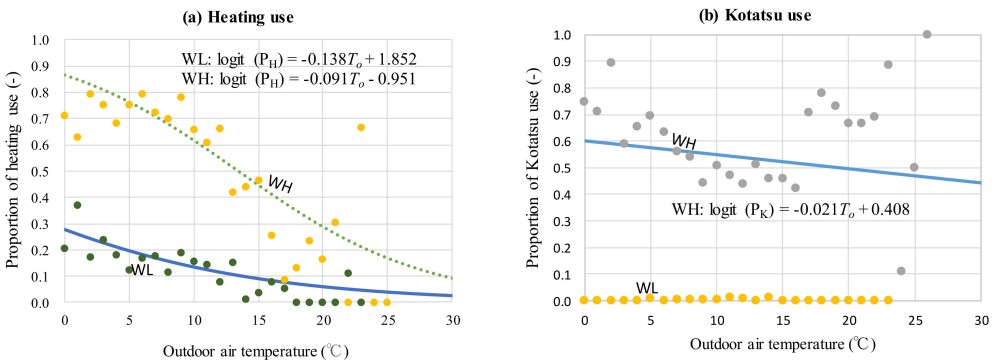

**Figure 12.** Relationship between heating use and outdoor air temperature for higher and lower temperature groups: (**a**) Heating use and (**b**) Kotatsu use.

The use of traditional heating device, 'Kotatsu', was observed for both groups because Kotatsu is part of winter culture for Japanese residents. As shown in Figure 12b, the proportion of Kotatsu use for WH is high; it was equal to 0.60 when the outdoor air temperature is 0 °C, while the use of Kotatsu for WL was almost none.

## 4. Discussion

In general consideration, occupants become fully dependent on the use of mechanical heating and cooling systems for better indoor thermal environments, especially when they live in smart houses where energy use can be controlled intelligently by automatic systems. The studied condominium is HEMS managed, which provides occupants with information about the outdoor thermal environment and the amount of energy used for a particular indoor device. Therefore, occupants living with such a system should be able to control and adjust the indoor thermal environment to their preference with the use of mechanical heating and cooling.

The results of this study show that the occupants were not dependent on mechanical system alone, indicated by temperature variations not being in narrow ranges. The occupants of both groups living in this condominium adapt to the fluctuating indoor thermal environment by various other activities besides the use of mechanical systems. During summer, the indoor air temperature of the SH group was higher than the recommended indoor temperature for cooling in summer in Japan of 28 °C. The analysis confirmed that the adaptive behaviors of families differed from one to another.

If this knowledge is considered during the development and installation of HEMS in the future, the system may be more applicable and longer lasting. Better insulating materials are used in this condominium in comparison to Japanese conventional and detached houses so that there is less heat flow from inside to outside. This is probably the main reason why the indoor air temperature is much higher than in conventional detached houses in winter. This result looks similar to the findings of exergy research with respect to indoor thermal environment in winter [26].

The results and discussion of this study generalizes the knowledge that occupants adopt various behaviors even when living with a HEMS. Although, the use of heating and cooling is less, occupants are comfortable. Fanger's PMV and predicted percentage dissatisfied (PPD) models [28] exclude the adaptive process of occupants. A recently developed adaptive prediction mean vote (APMV) model [29] may be useful to predict the actual thermal sensation in dynamic environmental conditions. This model focuses on the effect of previous environmental and age factors. Adaptive thermal comfort standards need to be developed, focusing on the adaptive opportunities of the occupants. In general, it is considered that mechanical use of heating and cooling is the only way of adjusting thermal comfort but, in reality, it was found that occupants adopt various other behaviors to adjust their thermal environment. Some studies [4,5] proved that energy use and cost is affected by the behavior of people, especially in low income households [4]. However, a similar study of energy use patterns showed that occupants were also conscious of energy saving in smart living [30]. The indication of electricity use by HEMS might encourage occupants to think about the cost and energy saving. HEMS should be improved, providing alternative means for taking different adaptive opportunities [12]. HEMS should provide people with an opportunity to be freer in adapting various activities so that they can adjust their thermal comfort as they like. Additional software that provides information about the right situations for taking passive means such as window opening, or the use of cooling or heating, needs to be included with HEMS so that the system will be more efficient for use in the future.

## 5. Conclusions

From a series of analyses of data on thermal environment measurement and an occupant behavior survey conducted in a HEMS condominium in the Tokyo metropolitan area of Japan, we have found the following:

1. The indoor air temperature in the studied condominium was not similar according to season, floor or flat.

2. Even though the building was equipped with a HEMS, there was a large range of temperature variation in indoor air temperatures due to individual adaptive activities of the occupants, similar to that found in common detached houses and residential buildings.

3. The window opening, fan use, cooling and heating behaviors of families living with the HEMS for thermal comfort adjustments were not similar, even though they were living with the HEMS in the same condominium.

4. There was 2 °C difference between SH and SL in summer and a 4 °C difference between WH and WL in winter, which shows that the behavioral characteristics of these four groups were different even using the same HEMS.

**Author Contributions:** As the primary author, Rajan KC contributed to the data analysis, manuscript writing, and prepared the final draft of the manuscript. Hom Bahadur Rijal, Masanori Shukuya and Kazui Yoshida revised the content of the paper, and contributed to the editing and structuring the paper and advising on the data analysis.

**Funding:** This research was funded by Ministry of Land, Infrastructure, Transport and Tourism (MLIT) on the "Subsidy for Pioneering Project about $CO_2$ emission reduction in residential and commercial building II, 2013".

**Acknowledgments:** We would like to thank to all the occupants residing in Katsushima (studied building) for being cooperative during the survey.

**Conflicts of Interest:** The authors declare no conflict of interest

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
