# Peer review of "An Investigation of the Behavioral Characteristics of Higher- and Lower-Temperature Group Families in a Condominium Equipped with a HEMS System"

_buildings, doi:10.3390/buildings9010004_

Round 1

Reviewer 1 Report

Abstract, Line 19:  Change "rather" to "often."

Line 28: drop "and more"

Lines 31, 32: Whereas new types of energy supply, such as fuel cells used for  electricity generation for example, are being developed, they may cause more CO2 emission. 

Line 34:  ...in building certain types–particularly in residential buildings– ...

Line 36: concern

Lines 53,54: In Germany, the minister of economy authorized a bonus for those who could demonstrate less energy use than in the foregoing year.

Lines 56,57: Thermal comfort is "the satisfaction of the mind with the thermal environment and is assessed by subjective evaluation”

Lines 61, 62:  Mechanical heating and cooling are the major behavioural factors of occupants for regulating indoor  thermal comfort, which are of course the main reasons for indoor energy use. 

Line 63: living in cold regions 

Lines 71, 72:  in the buildings where occupants use mechanical heating and cooling to regulate indoor comfort tend to have a narrower temperature range 

Lines 81–83: if the occupants have more behavioral choices then the range of temperature may be made wider than the conventional guidelines. 

Lines 100, 101: the Meguro river flows along the northern and western sides. This area has a warm and temperate climate. 

Line 107: An eighteen-story condominium 

112:  180 mm thick RC and 40mm urethane sprayed insulation have been used in the walls with double-glazed windows. 

133:  The accuracy of the measurement sensors is slightly low allowing for inexpensive installation used at large scales.

140: An online questionnaire survey was conducted 

165–166:  The relative differences between indoor and outdoor air temperatures were greater in January and smaller in August. 

167-169: The highly insulating materials used in the building is one of the reasons for this temperature difference other than heating use, because a similar trend has been observed  with  flats using no heating use; i.e., FR mode. 

348: houses because the energy use can be controlled intelligently by automatic systems.

365 similar with the finding in e[x]nergy research (spelling) 

Author Response

Dear Reviewer

Thank you very much for providing comments for our manuscript. We have revised our manuscript according to the comments received from you. The manuscript has been checked by a native English speaker after the revision and some grammatical errors were corrected. The detail of the revisions made are mentioned in the attached file. Please, find the attachment.

Thank You

Reviewer 2 Report

This study deals with the inhabitants behaviour as function of the possibility to use a Home Energy Management System (HEMS).

The case study is an eighteen floors building with 356 flatss located in Shinagawa (South-West of Tokyo, Japan).

A questionnaire survey was conducted during one year (November 2015-October 2016) with the partnership of 47 families.

In addition, air temperature and relative humidity were recorded and the measured data wer matched with the voting times of the occupants.

A series of analysis was carried out on the data collected by using many parameters such as outdoor temperature, flats, floors, occupants clothing, door and window opening, fan use, cooling use and heating use.

This study is interesting as a whole and could deserve a publication in Buildings International Journal.

Nevertheless, I suggest amendments indicated below.

- The quality of most of the Figures must be improved

- Line 135 : « …with the high accuracy… »

- Figure 3 : how was computed the mean indoor air temprature ? Over the whole measurement period ? For the 47 flats ?

- Figure 4 : the indoor air temperature values correspond to what exactly (47 flats, each time step…),

- Figure 5 : 41 flats instead of 47. Please, why ?

- Line 248 : mix-mode stands for what exactly (colling use, fan use, heating use…) ?

- Line 249 : free-running stands for what exactly ?

Author Response

Dear Reviewer

Thank you very much for providing comments for the improvement of our manuscript. We have revised our manuscript according to your comments. The manuscript has been checked by a native English speaker after the revision and some grammatical errors were corrected. The detail of the revisions made are mentioned in the attached file. Please, find the attachment.

Thank You

Round 2

Reviewer 2 Report

The article was improved and now is acceptable for publication.